# Multiomics-Based Feature Extraction and Selection for the Prediction of Lung Cancer Survival

**DOI:** 10.3390/ijms25073661

**Published:** 2024-03-25

**Authors:** Roman Jaksik, Kamila Szumała, Khanh Ngoc Dinh, Jarosław Śmieja

**Affiliations:** 1Department of Systems Biology and Engineering, Silesian University of Technology, 44-100 Gliwice, Poland; jsmieja@polsl.pl; 2Faculty of Automatic Control, Electronics and Computer Science, Silesian University of Technology, 44-100 Gliwice, Poland; kamiszu970@student.polsl.pl; 3Irving Institute for Cancer Dynamics and Department of Statistics, Columbia University, New York, NY 10027, USA; knd2127@columbia.edu

**Keywords:** multiomics data, feature selection, feature extraction, machine learning, next-generation sequencing, lung cancer, survival prediction

## Abstract

Lung cancer is a global health challenge, hindered by delayed diagnosis and the disease’s complex molecular landscape. Accurate patient survival prediction is critical, motivating the exploration of various -omics datasets using machine learning methods. Leveraging multi-omics data, this study seeks to enhance the accuracy of survival prediction by proposing new feature extraction techniques combined with unbiased feature selection. Two lung adenocarcinoma multi-omics datasets, originating from the TCGA and CPTAC-3 projects, were employed for this purpose, emphasizing gene expression, methylation, and mutations as the most relevant data sources that provide features for the survival prediction models. Additionally, gene set aggregation was shown to be the most effective feature extraction method for mutation and copy number variation data. Using the TCGA dataset, we identified 32 molecular features that allowed the construction of a 2-year survival prediction model with an AUC of 0.839. The selected features were additionally tested on an independent CPTAC-3 dataset, achieving an AUC of 0.815 in nested cross-validation, which confirmed the robustness of the identified features.

## 1. Introduction

Lung cancer remains a significant global health challenge, with its prognosis often hindered by delayed diagnosis and the complex molecular landscape of the disease [1]. Despite significant strides in cancer research and treatment modalities, survival prediction for lung cancer patients is still far from accurate and thus remains an important area of research. Challenges in prediction arise from the intricate interplay of factors such as inadequate screening programs, late-stage symptom manifestation, and the inherent heterogeneity of non-small-cell lung cancer (NSCLC), which comprises the majority of cases [1].

In recent years, the advent of -omics technologies, encompassing genomics, transcriptomics, proteomics, and metabolomics, has provided a large number of molecular information on lung cancer [2,3,4,5]. These -omics approaches and especially those based on multi-omics studies [6,7], have enabled a deeper understanding of the molecular subtypes of NSCLC, notably adenocarcinoma (LUAD) and squamous cell carcinoma (LUSC), each shown to be characterized by distinct epidemiological [8] and molecular features [9,10].

To address the complexity of lung cancer prognosis, machine learning methods have emerged as powerful tools for predicting treatment outcomes and personalizing therapeutic interventions [11]. Early applications ranged from statistical decision theory [12] to neural networks [13], paving the way for more sophisticated algorithms like Bayesian networks [14], support vector machines [15], and deep learning methods [16]. All of these approaches are based on a variety of distinct data, ranging from medical images to molecular characteristics, some of which were reviewed for NSCLC in the context of prognosis prediction [17].

Initially, those methods were focused mostly on analyzing individual -omics data. However, in recent years, there has been a rising trend towards integrative approaches, known as multiomics, which involve combining different types of data [18,19]. Their primary aim is to uncover potential indicators for the risk of cancer development and prognosis of treatment outcomes. Reaching this goal would facilitate the identification of target groups for screening, establishing optimal screening intervals, conducting risk stratification [20,21,22], and ultimately determining the most promising modes of treatment [23].

Despite the progress in this area, there are evident challenges hindering the effective development and implementation of methods. They stem from incomplete data, incomparable datasets, low information content in data, and other factors, resulting in varying conclusions regarding the efficacy of the methods. For instance, there is a notable discrepancy in performance, measured via the area under the ROC curve (AUC), ranging from 0.767 to 0.94 for a single method across different research groups (refer to [24] for a comparative analysis). Feature selection, a critical aspect of machine learning, plays a pivotal role in enhancing the accuracy of predictive models by systematically identifying and retaining the most relevant attributes that contribute significantly to the model’s performance [25]. This process not only streamlines the model’s complexity but also aids in mitigating the risk of overfitting. Moreover, alongside feature selection, the complementary aspect of feature extraction assumes equal significance in refining the predictive capacity of models. Feature extraction involves distilling meaningful information from raw data, allowing for the extraction of essential patterns and characteristics that contribute to a more nuanced understanding of the underlying relationships within the dataset.

Feature extraction is particularly important when dealing with gene mutation data due to the inherent complexity and intricacies of genetic information. Whole-genome sequencing (WGS) and whole-exome sequencing (WES) have been flooding the research community with data for the last several years. While the data have significant potential to identify genetic sources of various diseases, including cancer, and predict treatment outcomes, they are rarely used to create machine learning models. This is primarily because gene mutation data are highly dimensional, with a vast number of genes and potential mutations, leading to a large feature space. This abundance of features can lead to the “curse of dimensionality”, making it challenging to identify relevant patterns and discriminate between different classes effectively.

Moreover, the biological significance of gene mutations can vary widely, and not all mutations may contribute equally to the classification task. Prioritizing and selecting informative features becomes critical, requiring a deep understanding of the biological context and the specific relationships between gene mutations and the targeted outcomes. Another complicating factor is the presence of interactions and dependencies between genes. Genes often work in concert, and the status of other genes may alter the impact of a mutation in one gene. Capturing these interactions and integrating them into the feature extraction process adds additional complexity. Furthermore, the sparsity of mutation data poses a challenge. In many cases, the majority of genes may not exhibit mutations, leading to a sparse and imbalanced dataset. Traditional machine learning feature methods may struggle to handle such sparsity and may not effectively capture the subtle patterns associated with rare mutations.

Genomic feature extraction focuses either on local mutations or, more commonly, on their distribution in the genome. In the latter case, non-negative matrix factorization (NMF) is the most frequently used method, as it facilitates biological interpretation of the findings [26]. There is an abundance of NMF-based signature extraction tools, including the very popular SigProfilerExtractor [27,28] and *signature.tools.lib* R package, which was recently used to study a large cancer cohort in search of new mutational signatures [29]. 

Deep neural networks are another tool employed in this family of problems. In [30], a deep neural network Mutation-Attention (MuAt) has been proposed as a tool to learn representations of simple and complex somatic alterations to predict tumor types and subtypes. Its main idea was to create a feature space defined in terms of similarity to other mutations in the same tumor and subsequently use it to compute probabilities over the tumor type labels. While deep learning has a huge potential for the extraction of relevant genomic features [31], it requires vast amounts of data for the network to be created. This makes its application difficult or even infeasible in studies that include small numbers of cases.

Focusing on local, individual mutations does not allow the capture of structural variants (SVs) that affect larger fractions of the genome. Therefore, distribution-based feature extraction (DBFE) approaches have been proposed, considering frequencies of different lengths of SVs, including constant binning, quantile binning, clustering, and kernel density estimation. While the results concerning breast cancer datasets are very convincing, the AUC index for lung cancer is relatively low, ranging from 0.425 to 0.650 depending on the dataset and algorithm used [32]. Some of the methods are tailored to particular forms of SVs. For example, a feature extraction and selection algorithm based on statistically significant short tandem repeats (STRs)-based features and their respective enrichment patterns has been proposed in [33]. Mutation frequencies are employed using archetypal analysis to identify profiles representing extreme patterns in data [34].

Yet, another approach consists of treating mutational signatures as multinomial probability vectors representing expected proportions of mutations in each of the mutation classes. Then, tumor mutational spectra are analyzed, represented by mutation-class-wise sums of mutations caused by several mutational processes. It has been implemented in the mSigHdp software, using hierarchical Dirichlet process mixture models to discover mutational signatures [35].

The main idea behind signature extraction is to determine the influence of genotoxic substances and aberrations in specific cellular processes that leave a permanent mark on the genome rather than focusing on individual somatic mutations that are difficult to identify reliably. For example, a high frequency of deletions with microhomology sequences can indicate an impaired homologous recombination-based DNA repair process [36], a positive prognostic marker for overall survival in ovarian cancer [37]. However, while this approach has a huge prognostic potential, it may not be applicable to all prediction problems.

This study aims to address the challenges associated with feature extraction and selection by exploring the potential of machine learning in conjunction with multi-omics data to improve the prediction of lung cancer survival. Our study focuses on identifying the most relevant -omics datasets or their combinations and devising optimal feature selection strategies, especially in the context of sparse predictor matrices.

## 2. Results

### 2.1. Feature Evaluation Strategy

The primary goal of this work is to extract and select the most useful molecular features for the prediction of lung cancer survival. For this purpose, we use two distinct lung adenocarcinoma multiomics datasets obtained from the Cancer Genome Atlas (TCGA) and Clinical Proteomic Tumor Analysis Consortium 3 (CPTAC-3). In total, the study utilized five -omics data types: (1) somatic mutations obtained in whole-exome sequencing (WES), and (2) gene expression levels from the total RNA sequencing (RNA-seq), microRNA expression levels, methylation levels, and copy number variation (CNV).

The TCGA dataset includes more cases (N = 267) and -omics data types, and for this reason, it was used to identify relevant features. CPTAC-3 data (N = 96) were used as a hold-out set to test the prediction capabilities of selected features. Since our goal is not to create the best classifier, we only transferred information on extracted molecular features between the datasets and not the model itself. For the same reason, we did not include any features derived from clinical data, like the TNM staging system [17], focusing on molecular data instead. For a complete list of TCGA and CTPAC-3 cases selected for this study, see Appendix A.

We tested various feature extraction techniques using Lasso variable importance and Boruta feature ranking. Lasso was used to evaluate entire groups of features at a single -omics level, while Boruta was used to evaluate the contribution of individual features without information on their association with specific feature extraction methods and -omics datasets. Lasso was only used to compare different feature extraction methods and to show which of the -omic datasets provides the best predictor sets. We did not rank the predictors using Lasso for a combined -omic dataset since those results would be likely biased by the differences in measurement scales of individual datasets. When features have different measurement scales, their coefficients in the Lasso regression model are penalized differently. Consequently, features with larger scales may appear more important in the model simply because they have larger coefficients, even if they may not be more informative. To address this issue, we specifically opted for the Boruta feature selection technique, which is not affected by the measurement scale.

To evaluate the feature sets on both datasets, we used nested cross-validation comprised of an outer 10-fold cross-validation (CV) loop used to evaluate models on various subsets of the data (for a schematic representation of the process, see Figure 1). The parameters were independently selected in the inner leave-one-out cross-validation (LOOCV) to avoid overfitting the parameters to the specific training and testing subsets. The reason we used LOOCV instead of 10-fold CV in the inner loop was because of the limited number of available cases, especially the 12 non-survivors in the CPTAC-3 set, which would even further limit the number of cases used to create the model, especially when combined with sample group balancing. A random forest classifier was used to create the prediction model, which did not reduce the selected feature space through the built-in feature selection. Further details on the methodology utilized are available in the Section 4.

### 2.2. Feature Extraction

Feature extraction is a necessity when dealing with structural changes in the DNA, which in raw form provide a very sparse predictor matrix. This problem concerns two out of five -omics datasets that are used in this study: somatic mutations and copy number variation (CNV). Both methods provide rarely identical variants between multiple samples, making it difficult to define numerical features that make up good predictors. For this reason, we tested various feature extraction techniques with the goal of selecting the one that provided the most useful predictors. In this study, we present the results obtained for the following selected approaches:**Region overlap aggregation**—Combining altered regions based on a partial overlap. This is achieved by dividing the genomic regions representing alterations into smaller sections that are shared by at least two samples. Since the mutation set contains primarily single nucleotide variants (SNVs), this method is applicable only to the CNV set. Each smaller region is assigned a distinct copy number value, represented as a segment mean variable defined with log2(CN/2), where CN represents the absolute copy number. It should be emphasized that although this method involves segmenting the genome across all samples, not every genomic region is assigned a numerical value. Sections lacking alterations in at least one sample are omitted;**Gene level aggregation**—Aggregating mutations and copy number changes at the gene level by counting the total number of mutations associated with each gene. This approach is similar to the region overlap; however, the genome is divided based on gene coordinates, irrespective of the size of the altered regions. We then assign 1 to each gene that contains at least one protein-altering mutation and 0 if no such changes were detected. In the case of CNV, we assign the CN to the region in which the gene is located. We use the altered copy number value even if the region does not include the entire gene. Unfortunately, under certain conditions, particularly when the total length of copy number-altered regions is minimal, this method can also yield a sparse feature matrix. Additionally, it is important to recognize that genes situated in close proximity are likely to show a high correlation with the assigned values;**Gene set aggregation**—Grouping genes into biological pathways and aggregating mutations at the pathway level. This method provides a higher-level view of the functional impact of mutations and CNVs. This is achieved by further aggregating mutations and CNV statistics obtained for individual genes across specific gene sets that represent genes involved in the same signaling process, having similar functions, or showing altered expression levels as a result of specific stimuli. By adopting this method, both high-correlation and low-variance issues can be avoided. Furthermore, it has the potential to decrease the overall number of features, especially in scenarios with an exceptionally high number of aggregated regions;**Mutation statistics**—Calculating the number of specific mutation types, which are aggregated using three methods: SBS, DBS, and ID. In SBS (single-base substitutions), we calculate the number of variants depending on the reference and observed nucleotide and its context, e.g., G[T>C]T indicates a T-to-C mutation in a context of G and T on both sides (there are 96 types in total). DBS are the frequencies of double-base substitutions generated concurrently modifying two consecutive nucleotide bases, e.g., AC>GA (78 types). IDs are small indels (insertions and deletions) aggregated into 83 different types, based on their length, number and length of repeat units, and length of microhomologous sequences (sequence found at the edge of the deleted sequence and in the reference genome right next to it);**Mutation signatures**—Exploring the previous mutation statistics to identify mutational signatures associated with specific biological processes or exposures. Each of the mutation statistic groups (SBS, DBS, and ID) was used to calculate the association of each sample with individual signatures from each group;**PCA**—Principal component analysis (PCA) constitutes a distinct category within the spectrum of feature extraction methods, allowing the condensation of the feature space to match the number of tested samples. PCA is widely used in machine learning to maximize the variance or dispersion along the newly delineated variables.

Figure 2 illustrates the area under the curve (AUC) statistics for a specific number of predictors obtained using various feature extraction techniques. The AUC statistics were obtained by selecting a specific number of features using *Lasso* regression, which were used to classify TCGA patients into survivors (at least two years post-initial diagnosis) and non-survivors. The AUC serves as a metric quantifying a classifier’s capability to differentiate between classes and serves as a concise summary of the receiver operating characteristic (ROC) curve. A higher AUC value indicates a more effective classifier.

The gene set aggregation in both datasets showed the highest performance, with an AUC of ~0.75 and ~0.7 for 25 mutations and CNV features, respectively. This is consistent with our previous results using preselected features [38].

Figure 2 shows only selected aggregation types and their variants. We tried multiple other variants, e.g., by calculating frequencies instead of counts or MDS (multi-dimensional scaling) and NMF (non-negative matrix factorization) in addition to PCA; however, all yielded worse results and were omitted for plot clarity.

### 2.3. Evaluation of Prediction Capabilities of Each Dataset

One of the main objectives of this work is to determine which of the -omics datasets provides the most useful predictors for LUAD patient survival. To achieve this goal, we selected the best feature extraction techniques tested in Section 2.1 and conducted a multi-omics comparison, evaluating the feature usefulness not only for a 2-year survival prediction but additionally for the 5-year time period. Figure 3 shows that methylation and mRNA studies in both cases provide the most useful predictors, allowing AUC to exceed 0.8. Additionally, mutation data also show a similarly high AUC, though only in the case of 2-year survival. This suggests that somatic mutations have a smaller impact on survival exceeding 2 years; however, one also has to take into consideration that the 5-year survival includes a smaller number of samples, resulting from insufficient follow-up information post-diagnosis, leading to higher data loss (see Section 4 for details).

For the CNV dataset, we were unable to obtain 100 features for the testing since most of them were dropped by correlation-based filtering that removed all features with identical values across all samples. Since the 5-year survival is based on a lower number of samples compared to 2-year survival (185 and 267, respectively), it affected the correlation filtering, allowing more features to pass.

### 2.4. Variable Importance Assessment

In Section 2.3, we employed Lasso regression to assess the predictive potential of various -omics datasets. However, we found it performs poorly when multiple different -omics datasets are combined into a single feature matrix. Features from a single dataset were favored, and surprisingly, the AUC estimates were worse than those obtained for individual -omics datasets. This likely results from a significant variability in the measurement scales utilized by each -omic dataset, despite our attempts to standardize them. Consequently, a classifier built on such predictors fails to harness the potential of incorporating diverse datasets. We turned to the Boruta feature ranking and selection method to address this limitation. Boruta is based on the random forests algorithm, which is an ensemble of decision trees that are scale-independent. For this reason, various -omic datasets that utilize significantly distinct measurement scales can be combined without any additional transformations. The primary advantage of Boruta lies in its ability to decisively identify the importance, or lack thereof, of each variable for a specific classification problem. In this study, each variable is assessed individually rather than focusing on 1–100 best features, as in Section 2.2 and Section 2.3.

We conducted Boruta analysis using combined feature matrixes obtained with all feature extraction methods shown in Figure 2. The method was executed 10,000 times, providing as many variable importance scores for each of the attributes as possible. The analysis revealed that 40 features were confirmed to be of high importance, and an additional 4 features were classified as tentative (Figure 4a). The analysis revealed that only features originating from the methylation and mRNA datasets were selected, with one of the CNV features classified as tentative. The confirmed features were further ranked based on the average importance scores (Figure 4b), showing that none of the two methods is preferable and that they both provide variables of high importance, with cg27326750 probe methylation levels and RAET1E gene expression intensity being the most important two (details are available in the Appendix A).

Features highlighted in Figure 4b were further used to evaluate their classification potential based on a random forest classifier. Using 10-fold cross-validation repeated 100 times, we obtained an AUC of 0.839 for the TCGA dataset (Figure 5a). The selected features were evaluated on the same dataset used to extract them; for this reason, the obtained AUC value might be overly optimistic due to information leaks. To test those features on an independent dataset, we used the CPTAC-3 data, which included an additional 96 cases. Unfortunately, the CPTAC-3 dataset is based on a different methylation microarray, in which four of the selected CpG probes were unavailable, and for an additional four, the data were missing in the majority of samples. This required reducing our feature set from 40 to 32, for which we retested the classifier, first on the TCGA data (AUC = 0.83) and later on the CPTAC-3 samples (AUC = 0.815). The discarded features are marked in Figure 4b with red asterisks. The CPTAC-3 validation was, however, conducted only for 85 out of 96 individuals since 11 cases were discarded because they were missing methylation data for the majority of the selected probes in the 32 feature set.

Figure 5b is a so-called Beeswarm plot, showing SHAP (SHapley Additive exPlanations) values that measure the input features’ contribution to individual model predictions [39]. Each dot represents one patient. Positive SHAP values increase the probability that the model will predict the positive class (patient dies within 2 years of diagnosis), which, combined with the values of the features represented through the color scale, allows us to conclude what values of each feature are associated with an increased/decreased patient survival. The plot reveals that while most of the features selected using the TCGA dataset are also good predictors in the CPTAC-3 set, some of them show an opposite contribution to the model (high value is more important for the positive class or the other way around), e.g., cg10441898. This is likely a result of the small number of cases in each dataset; however, the differences between experimental strategies used in both datasets might also be a significant contributing factor.

The mean absolute SHAP values (available in Appendix A) are additionally an indicator of the variable importance. The three most important variables for the CPTAC-3 model are RAET1E, cg27326750, and SOX9-AS1. RAET1E (also known as ULBP4) is a protein-coding gene involved in the immune response, which was previously associated with poor prognosis in ovarian [40], bladder [41], and nasopharyngeal cancers [42]. SOX9-AS1 is a long non-coding gene (lncRNA) located upstream from the SOX9 transcription factor that was shown to be associated with poor prognosis in liver cancers: cholangiocarcinoma [43] and hepatocellular carcinoma [44]. cg27326750 is a methylation probe located within the PRM2 protein-coding gene, which has not been linked with cancer yet.

## 3. Discussion

In this study, we tested the potential of various -omics datasets for the prediction of 2-year survival of lung cancer patients. We showed that gene expression changes (RNA-seq), methylation, and mutation information provide significantly better predictors than CNV and microRNA expression levels, a result consistent for both 2-year and 5-year patient survival. This observation is important to reduce costs associated with testing lung cancer patients. Eliminating two families of tests needed for prognosis should significantly reduce the financial burden imposed on healthcare systems, taking into account the prevalence of lung cancer. It is particularly important that microRNA measurements, still challenging with respect to their accuracy (see, e.g., [45]), were found to be less important. Additionally, we successfully showcased the classification performance achieved through various feature extraction methods on the mutation and CNV datasets. Our conclusion emphasizes that gene set aggregation leads to the best classification results.

Gene set aggregation addresses the problem of sparse predictor matrices, especially in the case of mutations that contain primarily zeros, which indicate that a particular gene was not mutated in an individual’s genome. However, it is a known fact that mutations in multiple genes can have a similar effect on the phenotype, primarily in genes that share a common function or are dependent on each other through the regulatory processes. One of the most relevant examples of lung cancer are the EGFR and KRAS oncogenes. Mutations in those genes are mutually exclusive in lung adenocarcinoma [46], and both impact the signaling pathways that regulate cell growth [46]. Each of them individually shows no association with 2-year patient survival; however, after aggregating them across all of the TCGA LUAD cases, the association becomes significant, as tested with the Pearson chi-squared test (*p*-value = 0.05). The gene set aggregation technique takes this to another level by considering all known gene associations based on genomic location, function, or regulation (35,774 distinct gene sets).

In this work, we used all of the available features, unlike in our previous study [38], where pre-selection was utilized, to extract the most relevant multi-omics features. The feature set selected using 267 TCGA cases was independently evaluated on 85 CPTAC-3 samples, allowing the achievement of an AUC of 0.815 in a nested cross-validation, based on a random forest classifier. The AUC value obtained for the TCGA data used to select the features was 0.839. This is a lower value than the one we obtained in our previous study [38], which was 0.851. However, here, we employed a much more sophisticated cross-validation, which included model tuning, and we also performed the Boruta feature selection on a different set of features, which led to a slightly different final feature set.

Using the available literature, it is generally very difficult to compare various survival prediction models based on molecular data. The main reasons are differences in the definition of survival (2 or 5 years), cases used (that may represent only specific patient subgroups with distinct treatment strategies or specific tumor grades), and even evaluation methods. Some studies report the AUC values [47,48], some focus on accuracy [49,50], while others maximize the C-index [51,52]. The reported AUC values vary between 0.68 and 0.84, as summarized in [24]; however, the highest 0.84 was obtained only after including features derived from the computed tomography images [48]. Taking into account the described differences, the AUC values reported in this work show a considerable improvement over previously reported values. This is especially evident in a case of studies based on molecular data alone, like the AUC = 0.68 in [48] and 0.723 in [53]. Also, the AUC value that we obtained for an intendant dataset (0.815) is considerably higher than the AUC of 0.729, reported in [47] for an independent dataset that included both gene expression and clinical data.

The relatively high AUC values obtained in this work, 0.839 and 0.815 for the 2-year survival prediction models in the TCGA and CPTAC-3 datasets, respectively, highlight the robust predictive ability of selected lung cancer characteristics in terms of survival time. When timing and accurate prognostic information are most important for lung cancer, achieving AUCs of 0.839 and 0.815 is clinically significant, especially since the model can be potentially tuned to achieve >0.8 sensitivity (as shown in Figure 5a) with >0.7 specificity. These results suggest that our predictive models have the potential to identify the majority of individuals at increased risk of poor outcomes. Such predictive accuracy is sufficient to support clinical decision-making, such as adjusting the treatment strategy.

The random forest classifier used to perform the analysis was selected for its robustness and versatility, particularly in the context of our multi-omic study. One of its key advantages lies in its ability to handle diverse types of data without additional pre-processing or normalization, making it well-suited for integrating multiple omics datasets with varying measurement scales. Moreover, the ensemble learning approach in the random forest, which aggregates the results from multiple decision trees, improves the model’s forecasting accuracy and overall generalization ability. This is especially important in the context of multi-omics data, where interactions between different factors within and across omics can be complicated and nonlinear.

It is worth noting that in this study, we only transferred the information about selected features between both TCGA and CPTAC-3 datasets. The classifier trained on TCGA data performed poorly on the CPTAC-3 set, which was expected due to significant differences between methods used to obtain both datasets and differences in the pre-processing pipelines. Also, we applied an inter-sample standardization procedure (DESeq2) for the RNA sequencing pipeline, which was conducted independently for both datasets. In order to reliably test the model, the two independent datasets, TCGA and CPTAC-3, changes in the pre-processing pipelines would be necessary to match the methodology, along with additional standardization to mitigate batch effects, particularly within the RNA sequencing datasets. However, batch effect correction would be difficult in this setting primarily due to the use of various measurement platforms. There are many TCGA features that are not available in the CPTAC-3 dataset and vice versa. For this reason, rather than directly combining the datasets, we concentrated solely on the predictors within each dataset. The goal of this study was not to provide the best classifier but rather to evaluate feature extraction methods and select the best set of predictors. Consequently, validating the TCGA model on the CPTAC-3 data was not deemed necessary for this purpose. It is also worth noting that the CPTAC-3 dataset was chosen after the features were selected using the TCGA data, based on the availability of both methylation and gene expression data for the same cohort of lung adenocarcinoma patients. The limited availability of other methods, especially microRNA expression levels, had no impact on the feature selection process. Given the variability in the timing of the last follow-up, we chose to concentrate on 2-year rather than 5-year survival. Cases where the last follow-up occurred before 2 years, and the patient was alive at that point, were excluded. In such instances, it was impossible to ascertain whether the actual survival time reached at least two years from the initial diagnosis. Such cases had to be removed from the analysis, which for 5-year survival resulted in a significant data loss.

While the relatively small number of cases used for the model creation and testing (N = 267 + 85 for both datasets) is a considerable limitation of this study, to our knowledge, this is the largest such study performed on lung adenocarcinoma. This stands out particularly when contrasted with other multi-omics classification approaches targeting lung cancer [48,49,54,55,56], where the case numbers ranged from 28 to 168.

As the work focused on -omics-based features, other potential confounding factors or biases that may affect the survival prediction have not been included in our analysis. Of these, smoking status has been long known to be an important predictive factor, and one should expect that it will remain so, in addition to features indicated by -omics analysis. It is also clear that comorbidities may highly influence the survival of lung cancer patients. However, how to include them in predicting patients’ survival remains an open question, as comorbidity profiles differ across sex and race [57]. An interesting simulation study can be found in [58], showing that treatments for lung cancer that maximized quality-adjusted life expectancy differed by comorbidity status, age, sex, and tumor size and, therefore, less invasive treatments might be recommendable to some patient groups, particularly those with multiple major comorbid conditions. Sex itself is an intriguing factor, as recent publications reported that while there are differences between treatment type, rates of endocrinologic complications after immunotherapy use, and rates of psychological disorders, sex did not prove to be an independent factor associated with survival [59]. Age has also been considered an important prognostic factor [60,61,62]; it appears to be more a result of a health system approach to the treatment of elderly patients than an intrinsic feature of cancer. For those reasons, including clinical information and molecular data could increase the accuracy of the prediction models created in this project.

The study could also benefit from an additional stratification of patients based on the selected treatment. This, however, would require incorporating additional datasets to increase the number of cases. It is also worth noting that the genomic features that we used did not include structural variants (SVs) [32], and some more complex events like microsatellite instability (MSI) [63], telomere content [64], or chromothripsis [65], which could also provide useful predictors. The inclusion of additional genomic features has just recently become possible due to the publication of whole-genome sequencing data for the majority of TCGA cases.

Our study focuses on identifying the most relevant -omics datasets or their combinations and selecting the optimal feature extraction strategies, especially in the context of sparse predictor matrices. This process involved many challenges associated with the feature selection and extraction processes and data quality, which likely includes patients whose survival status might not be related to the cancer itself. This is likely the main reason why survival classifiers are generally characterized by low AUC values. Despite these challenges, we strive to contribute to the advancement of precision oncology by providing a relatively small feature set that might be used to create more accurate prognostic tools, which would not require high throughput measurement methods like next-generation sequencing or oligonucleotide microarrays.

## 4. Materials and Methods

### 4.1. Multiomics Data

Pre-processed genomic and transcriptomic data were downloaded through the GDC Data Portal at https://portal.gdc.cancer.gov (accessed on 24 January 2023). The dataset included 518 lung adenocarcinoma (LUAD) patients from the Cancer Genome Atlas (TCGA) project and 229 patients from the Clinical Proteomic Tumor Analysis Consortium 3 (CPTAC-3). The following types of data were downloaded:Gene expression levels derived from total RNA sequencing (RNA-seq) in the form of raw read counts assigned to individual genes;microRNA expression levels derived from RNA sequencing (miRNA-seq) in the form of raw read counts assigned to individual miRNAs;Locations of somatic mutations (single nucleotide variants and short indels) obtained in a whole-exome sequencing experiment (WES) in the form of genomic coordinates (for the GRCh38 genome version) with supporting annotation data;DNA methylation levels acquired via Illumina Infinium HumanMethylation450 (TCGA) or Methylation Epic (CPTAC-3) BeadChip microarrays, expressed as values within a range of 0 to 1;Copy number variation (CNV) data acquired using Affymetrix SNP 6.0 microarrays, presented as genomic intervals (GRCh38 reference genome), along with segment mean statistics (log2(CN/2), where CN represents the observed copy number of a particular region).

We also gathered clinical information for each patient, including the survival time represented in days from the initial diagnosis. The patients were categorized into two groups: those who died within two years of diagnosis and those who survived beyond that timeframe. We excluded from the study all instances where data from any of the five TCGA platforms (N = 78) or two CPTAC-3 platforms (N = 1) were unavailable.

We additionally excluded cases for which determining the 2-year survival was not feasible, specifically when the last follow-up occurred earlier than two years from diagnosis, and the patient was alive at that time. We also excluded cases in which the patient died due to reasons other than cancer, which, unfortunately, was possible only in the CPTAC-3 project since information on the cause of death was unavailable for the TCGA patients. This filtration, based on the annotation information, was the most significant contributor to the reduction of the dataset size by 173 cases in TCGA and 133 in the CPTAC-3 datasets. As a result, 267 TCGA and 96 CPTAC-3 cases were considered for further analysis, comprising 178 and 84 identified as survivors and 89 and 12 non-survivors in TCGA and CPTAC-3 datasets, respectively. For the 5-year survival study conducted using the TCGA dataset, the number of cases was further decreased to a total of 185 samples, including 140 survivors and 45 non-survivors, due to missing follow-up information.

### 4.2. Feature Definition

Data acquired with the RNA-seq method provided 60,483 features representing expression levels of coding and non-coding genes. Features that showed less than 10 reads in all samples were removed from the study. The data were then standardized using the *DESeq2* method ver.1.40.2 [66]. An identical procedure was applied to the microRNA-seq data, providing information on 1881 features associated with the expression levels of individual microRNAs.

Data from the WES method provided 229,439 unique missense somatic mutations, aggregated into 18,409 genes and later into 35,774 gene sets defined in the Molecular Signatures Database (MSigDB) v.7.5 [67]. We used the entire MSigDB, which, among others, included ontology gene sets (genes annotated by the same Gene Ontology term), signaling pathway gene sets (based on KEGG, Reactome, BioCarta, and other databases), regulatory target gene sets (potential targets of regulation using specific transcription factors or microRNAs) and oncogenic signature gene sets (signatures of cellular pathways which are often dis-regulated in cancer). Mutation signatures (SBS, DBS, and ID) were extracted using the *signature.tools.lib* R package ver.2.4.2 [29], based on ver.3.4 of the COSMIC Mutational Signatures database [68].

Data from the copy number variation (CNV) study provided a total of 132,327 genomic ranges with altered copy numbers across all samples. Since the ranges are very rarely identical, we used the *CNregions* function from the iClusterPlus R package ver.1.36.1 to aggregate them, using partial overlap information (*frac.overlap* = 0.9). This resulted in 3122 features across all samples, which we further associated with gene and gene set definitions, similarly as for the mutation data.

The methylation dataset contained a total of 485,577 genomic locations (CpG islands) for which methylation levels (represented as beta values) were measured across all samples.

### 4.3. Variable Importance Study

Variable importance study was conducted using two distinct approaches, least absolute shrinkage and selection operator (Lasso) regression, using the *glmnet* R package ver.4.1-8 [69], and, additionally, the Boruta feature ranking and selection algorithm, which is based on the random forest approach, implemented in the *Boruta* R library ver.8.0.0 [70]. Lasso was executed for each of the -omics datasets independently, in a 10-fold cross-validation, through the *cv.glmnet* function, with the following parameters: *family =* ‘*binomial*’, *nfolds* = 10, *alpha* = 1, *standardize* = *TRUE*, and *type.measure* = ‘*auc*’. Boruta was executed for the combined -omic datasets, using a maximum of 10,000 runs (*Boruta* function with the following parameters: *maxRuns* = 10,000, *getImp* = *getImpRfZ*). In both cases, features with variance equal to zero and highly correlated features were excluded from the study (correlation coefficient > 0.9) using the *findCorrelation* function from the *caret* R package ver.6.0-94 [71].

### 4.4. Classification of Data

The process of building a predictive model was performed in the R environment using *caret* ver.6.0-94 [71] and *nestedcv* ver.0.7.0 [72] R packages. The models were obtained using the random forest method. Other approaches, including SVM, LDA, GLM, and PLS, were also tested; however, those performed worse in terms of the AUC value obtained for the TCGA dataset. This was consistent with the results that we obtained in our previous study [38].

The quality of the models was evaluated using nested cross-validation [72], which included an outer 10-fold cross-validation used to determine model performance and inner leave-one-out cross-validation to tune the models. Since both used datasets were unbalanced, we applied a down-sampling process for case selection in a single cross-validation loop, which reduced the number of cases from a larger group (in this case, ‘survivors’) so that both groups are represented by a comparable number of cases. As a result, the model is trained on a smaller number of cases, which affects its predictive ability but allows for a greater balance between sensitivity and specificity. The entire nested cross-validation was evaluated 100 times for each of the datasets. The nested cross-validation was executed through the *nestedcv::nestcv.train* function with the following parameters: *method =* ‘*rf*’, *tuneLength* = 10, *n_outer_folds* = 10, *metric* = ‘*ROC*’, *balance* = ‘*randomsample*’, *balance_options* = *list* (*minor* = 1, *major* = 0.3), *trControl* = *trControl*. The *trControl* variable was obtained through the *trainControl* function with the following parameters: *method* = ‘*LOOCV*’, *classProbs* = *TRUE*, *summaryFunction* = *twoClassSummary*, and *savePredictions* = *T*.

SHAP (SHapley Additive exPlanations) values were calculated using the *fastshap* ver.0.1.0 [73] R package, based on the *nestedcv* results. This was achieved through the *fastshap::explain* function, with the *nsim* = 10 parameter, evaluated independently for the 100 *nestcv.train* iterations. The beeswarm plots were created using the *nestedcv* [72] R package using SHAP values averaged across all 100 iterations. The ROC curves were created using *plotROC* R package ver.2.3.1 based on the results obtained in the outer cross-validation loop in all 100 iterations.

## Figures and Tables

**Figure 1 ijms-25-03661-f001:**
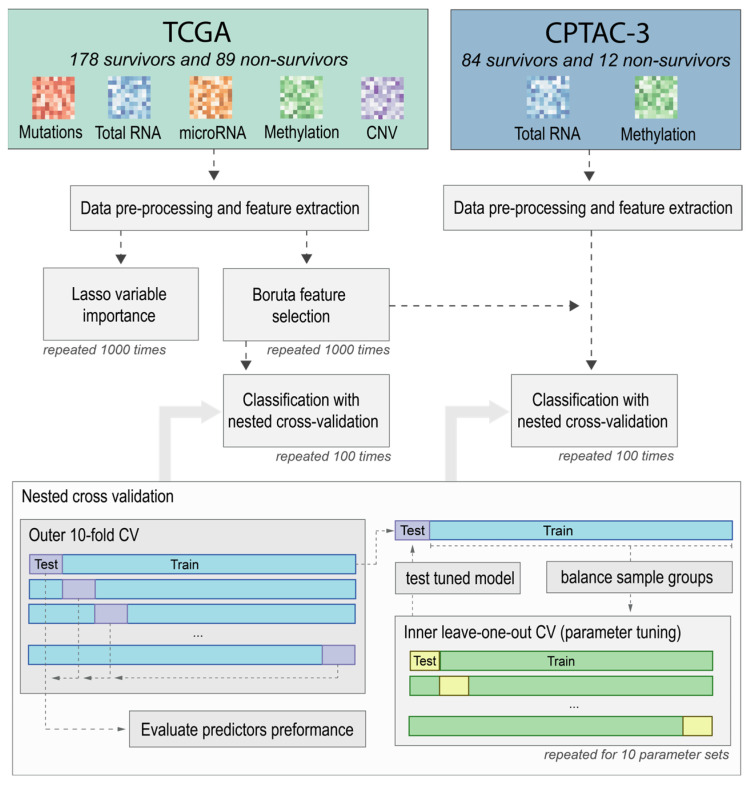
Schematic representation of feature evaluation strategy based on two lung adenocarcinoma patient cohorts: TCGA and CPTAC-3.

**Figure 2 ijms-25-03661-f002:**
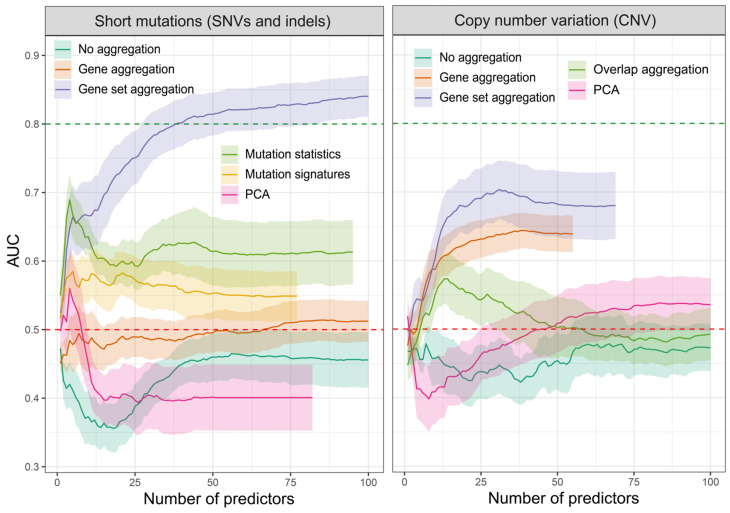
Area under the curve (AUC) for patient survival classification through the application of different dimensionality reduction methods on the mutation and CNV datasets. Statistics were derived from 10-fold Lasso regression, repeated for each individual dataset. Ribbons mark the standard deviation. The Red dashed line marks the AUC level of 0.5. AUC of 0.8, marked with a green dashed line, was exceeded only via the gene set aggregation method of short variants for a minimum of 38 features.

**Figure 3 ijms-25-03661-f003:**
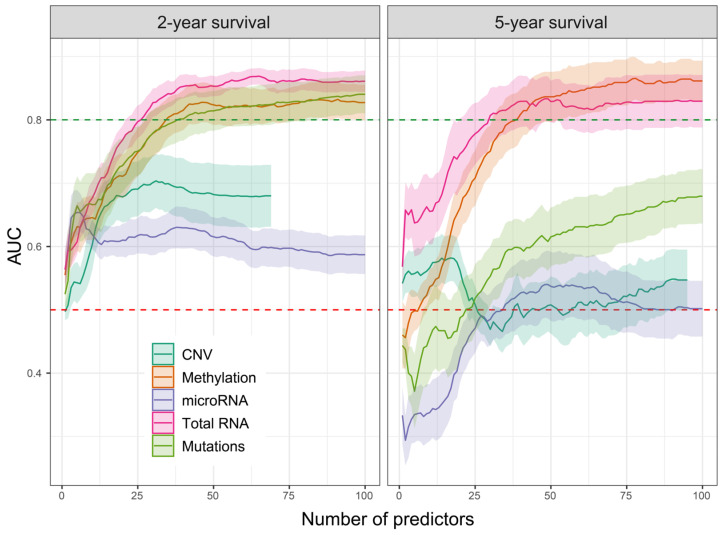
Area under the curve (AUC) for both 2-year and 5-year survival models, obtained using various –omics datasets. Statistics were derived from 10-fold Lasso regression, repeated for each individual dataset. Ribbons mark the standard deviation. The red dashed line marks the AUC level of 0.5. AUC of 0.8, marked with a green dashed line, was exceeded only by the total RNA and methylation datasets, in the case of both 2-year and 5-year survival and by the mutations dataset in the 2-year survival model.

**Figure 4 ijms-25-03661-f004:**
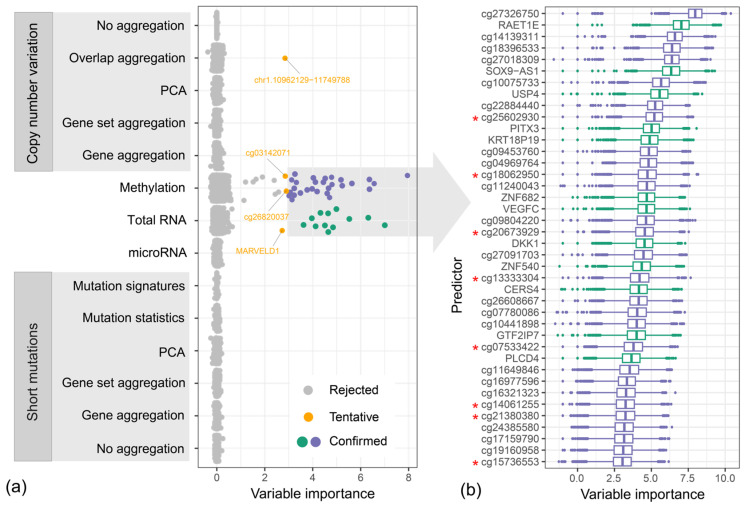
Boruta variable importance score for each of the predictors associated with individual methods and TCGA -omics datasets: (**a**) average importance score for each of the predictors, the confirmed variables are colored individually for methylation and mRNA datasets; (**b**) distribution of the variable importance scores for each of the 10,000 Boruta iterations. Red asterisks mark features that were later discarded from the validation, since they were not available in the CPTAC-3 dataset.

**Figure 5 ijms-25-03661-f005:**
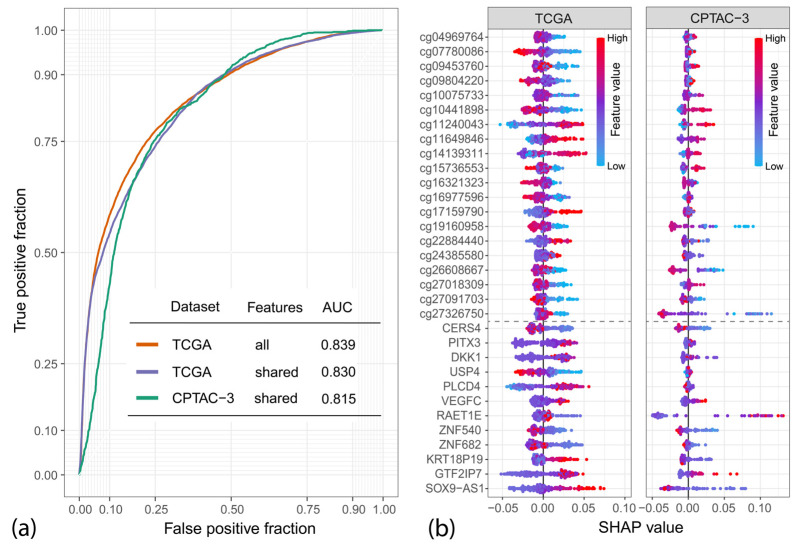
Results of the nested cross-validation on TCGA and CPTAC-3 datasets: (**a**) Receiver operating characteristic (ROC) curves obtained for all features selected using the TCGA dataset and their subset representing features available in both TCGA and CPTAC-3. The table contains the area under the ROC curve (AUC) values obtained for each variant; (**b**) Beeswarm plot for the SHAP (SHapley Additive exPlanations) values, showing the contribution of each variable to the individual predictions of the model. The color scale indicates the relative value of each feature; e.g., red indicates high methylation or expression level depending on the dataset. A dashed horizontal line separates the methylation (**top**) and gene expression features (**bottom**).

## Data Availability

All of the raw data used in this study are available through the GDC Data Portal at https://portal.gdc.cancer.gov (accessed on 24 January 2023). The code used to conduct the presented analysis is available on GitHub: https://github.com/rjaksik/LUAD_multiomic (accessed on 20 March 2024).

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
