# Peer review of "Multiomics-Based Feature Extraction and Selection for the Prediction of Lung Cancer Survival"

_ijms, 2024, doi:10.3390/ijms25073661_

Round 1
Reviewer 1 Report (New Reviewer)
Comments and Suggestions for Authors
This is a well designed bioinformatic study which established a survival prediction model of LUAD with a relative high AUC. The authors optimized several analyzing steps, including cross-validation, feature extraction, etc., and discussed throughly on the challenges of setting prediction models.
Minor suggestions:
There should be other databases with paired gene expression/variations/methylation information for lung adenocarcinoma. Could also validate the efficacy of this model in another independent cohort (real-world or public), since these three factors are important in the prediction model, and are also generally accessible.
Author Response
We agree that the addition of another dataset would be highly beneficial to this project. Unfortunately we were unable to find an additional dataset that would allow us to test the classification accuracy of the 32 selected features. The minimum requirements for that are:
- At least 20 lung adenocarcinoma patients included in the project
- Whole transcriptome sequencing or microarray data
- Genome-wise methylation profiles assessed by either Illumina microarrays or WGBS
- Clinical information that includes survival time, time of the last follow-up and preferably cause of death (to exclude non-cancer related deaths)
Besides CPTAC-3 and TCGA the two major LUAD projects are phs001169: Environment And Genetics in Lung cancer Etiology (EAGLE) and phs001179: Foundation Medicine Adult Cancer Clinical Dataset (FM-AD). However they are primarily focused on DNA and do not meet the above data availability requirements.
Reviewer 2 Report (New Reviewer)
Comments and Suggestions for Authors
The research article "Multiomics-based feature extraction and selection for the prediction of lung cancer survival" presents a comprehensive and methodologically sophisticated approach to improving lung cancer survival predictions through the integration of multi-omics data. The study's methodology encompasses several key components, including data acquisition, feature definition, variable importance study, and classification of data, employing advanced statistical and machine learning techniques to analyze and interpret the complex datasets. Below are critical analyses and questions on the paper's methodology and various sections:
1. The study utilized pre-processed genomic and transcriptomic data from TCGA and CPTAC-3, focusing on lung adenocarcinoma patients. Data types included RNA-seq, miRNA-seq, somatic mutations, DNA methylation levels, and CNV data, along with clinical information regarding survival time. Was there any consideration of potential batch effects, co-variates or differences in data processing pipelines between TCGA and CPTAC-3 datasets? How were these factors accounted for in the analysis to ensure comparability across datasets?
2. The researchers applied several feature extraction methods, including PCA, Lasso regression, and Boruta feature ranking, across different omics datasets. They assessed the predictive potential of various omics datasets both individually and in combination. Given the inherent differences in data types (e.g., continuous vs. categorical data in gene expression vs. mutation data), how were these differences reconciled in the feature extraction and selection process to avoid biasing the model towards one type of omics data?
3. The study used Lasso regression and the Boruta algorithm to identify important features. Lasso regression was employed for each omics dataset independently, while Boruta was used for combined omics datasets to rank features based on importance. The use of Lasso regression might favor features with larger effect sizes but potentially overlook features with smaller yet biologically significant effects. How does this selection bias impact the model's generalizability and interpretation?
4. The predictive model was built using the random forest method, evaluated through nested cross-validation. Other methods like SVM, LDA, GLM, and PLS were also tested but performed worse. How was the decision made to exclude other machine learning models from further consideration? Was there a thorough exploration of hyperparameter tuning or model architecture adjustments for these models?
5. The study tested the model's performance on independent datasets (TCGA and CPTAC-3), noting challenges in model transferability due to differences in data acquisition methods and preprocessing pipelines. How significant were the discrepancies in model performance between the training (TCGA) and validation (CPTAC-3) datasets, and what steps were taken to mitigate these differences? Was there any attempt to standardize preprocessing across datasets?
6. The authors acknowledge the exclusion of potential confounding factors such as smoking status and comorbidities from their analysis. Could the exclusion of these and other clinical factors significantly impact the survival prediction models' accuracy and applicability in a real-world clinical setting?
7.The study hints at the potential for further exploration of integrating additional omics data types and refining feature selection methods. What specific omics-data types or methodological improvements do the authors consider most promising for enhancing the predictive accuracy of lung cancer survival models?
8. Code and processed data needs to be present to researchers on GitHub or other platform for testing and reproductivity of the research.
The methodology employed in this study represents a significant advancement in the field of precision oncology, particularly in the context of lung cancer survival predictions. However, addressing the above questions could further refine the approach, enhance the interpretability of the results, and improve the models' clinical applicability.
Author Response
Please see the attachment.

Reviewer 3 Report (New Reviewer)
Comments and Suggestions for Authors
This is an interesting manuscript where address the challenges associated with feature extraction and selection by exploring the potential of machine learning in conjunction with multi-omics data to improve the prediction of lung cancer survival. The manuscript is well written and the content interesting but I only have 2 concerns prior to be considered for publication in IJMS:
1) Authors should improve the quality of figures
2) Authors should add statistical analyses performed in each figure
Author Response
We are very grateful for this kind review. We enhanced the figure quality making them fully vectorized. The main article now includes high quality rendered version (1000 dpi) and we additionally provided vector versions in the PDF format. We also corrected the figure descriptions in the text additionally highlighting some of the most important statistics.
Round 2
Reviewer 3 Report (New Reviewer)
Comments and Suggestions for Authors
Authors addressed all my concerns.
This manuscript is a resubmission of an earlier submission. The following is a list of the peer review reports and author responses from that submission.
Round 1
Reviewer 1 Report
Comments and Suggestions for Authors
The paper provides a comprehensive overview of the research, emphasizing the global challenge of lung cancer and the importance of accurate survival prediction. The use of multi-omics datasets from TCGA and CPTAC-3 projects, focusing on gene expression, methylation, and mutations, is highlighted. The study introduces new feature extraction techniques and unbiased feature selection methods to enhance survival prediction accuracy. Results demonstrate the identification of 32 molecular features for a 2-year survival prediction model with robust validation on an independent dataset.
1. Are there results for lung squamous cell carcinoma (LUSC) in addition to lung adenocarcinoma (LUAD) patients in the study?
2. Are there outcome values provided based on KRAS or EGFR mutations in the study?
3. Expand on the effectiveness of gene set aggregation as a feature extraction method for mutation and copy number variation data. Offer a brief explanation of why this method was chosen and how it improves feature extraction.
4. Consider providing a glimpse into the broader implications or future directions of the research.
5. In the results, discuss the clinical significance of achieving an AUC of 0.839 and 0.815 for the 2-year survival prediction models on the TCGA and CPTAC-3 datasets, respectively.
6. Provide a brief explanation of why the random forest classifier was chosen and discuss its advantages in the context of your study.
Comments on the Quality of English LanguageThe overall quality of the English language in the manuscript is quite good. The text is generally well-structured and communicates complex scientific concepts effectively.
The manuscript is well-written, and addressing these minor suggestions can further improve its clarity and accessibility to a broader audience.
Author Response
Thank you for providing this insightful review. Following your suggestions we significantly expanded the discussion adding 13 new references. Below are our detailed answers to each of the 6 points:
- In our study, we used data on patients with lung adenocarcinoma (LUAD) only and did not include results on lung squamous cell carcinoma (LUSC). LUAD and LUSC represent different subtypes of lung cancer, each with unique epidemiological and molecular characteristics. The divergence in their genetic and clinical profiles requires separate analyses, and any assumptions drawn from one subtype cannot be easily extrapolated to the other. We considered expanding our study by adding the LUSC data but ultimately we decided to focus on LUAD, since a separate analysis would likely reduce the clarity of the manuscript. Additionally, the number of samples, for which the information required by this study is available, was significantly lower in the LUSC dataset.
-
While KRAS and EGFR mutations were, among others, included in the study, they were not selected as the most relevant variables for the survival prediction. Both KRAS and EGFR are among the most frequently mutated genes in lung cancer, however, we found no association between them and 2-year patient survival in the TCGA LUAD dataset (p-values of 0.22 and 0.32 respectively, yielded by the Pearson's Chi-squared test).
-
The selected aggregation technique addresses the problem of sparse predictor matrices especially in a case of mutations that contain primarily zeros, which indicate that a particular gene was not mutated in the genome of an individual. It is, however, a known fact that mutations in multiple different genes can have a similar effect on the phenotype, primarily in genes that share a common function or are dependent on each other through the regulatory processes. EGFR and KRAS oncogenes mentioned above are widely recognized examples Mutations in these genes are known to be mutually exclusive in lung adenocarcinoma (Unni et al. 2015 Elife;4:e06907), and both are known to impact signaling pathways that regulate cell growth. By aggregating just those two mutations across the TCGA patients, the p-value from the Pearson's Chi-squared test, as conducted in previous comment, drops to 0.05.
Gene-set aggregation technique takes this to another level, by considering all known gene associations, based on genomic location, function or regulation. We cannot objectively determine which gene set is expected to be most relevant. However, the use of high performance computers allowed us to test all known associations (35 774 distinct gene sets), assessing their potential for the 2-year survival prediction. We added this justification to the discussion.
-
We are working on other, more complex feature extraction techniques for WGS data, which consider specific classes of variants and genomic events among others, e.g. indels with microhomologies, structural variants, chromothripsis events. The study could also benefit from an additional stratification of patients based on the selected treatment. This, however, would require incorporating additional datasets to increase the number of cases. We also believe that the features identified in this study can be already used to design new small scale tests, based on the expression of individual genes or methylation levels across specific regions of the genome. We extended the discussion to include this information.
-
The achieved area under the curve (AUC) values of 0.839 and 0.815 for the 2-year survival prediction models in the TCGA and CPTAC-3 datasets, respectively, highlight the robust predictive ability of selected lung cancer characteristics in terms of survival time. For lung cancer, where timing and accurate prognostic information are most important, achieving AUCs of 0.839 and 0.815 is clinically significant, especially since the model can be potentially tuned to achieve >0.8 sensitivity (as shown on Fig. 5a) with >0.7 specificity. These results suggest that our predictive models have the potential to identify the majority of individuals at increased risk of poor outcomes. Such predictive accuracy is sufficient to support clinical decision-making, for example to adjust the treatment strategy. We extended the discussion section to include this information.
-
The random forest classifier was selected for its robustness and versatility, particularly in the context of our multiomic study. One of its key advantages lies in its ability to handle diverse types of data without the need for additional preprocessing or normalization, making it well-suited for integrating multiple omics datasets that have varying measurement scales. Moreover, the ensemble learning approach used in the random forest, which aggregates the results from multiple decision trees, improves the model's forecasting accuracy and its overall generalization ability. This is especially important in the context of multi-omics data, where interactions between different factors within and across omics can be complicated and nonlinear. This information was included in the discussion.
Reviewer 2 Report
Comments and Suggestions for Authors
I am writing to provide feedback on your manuscript titled "Multiomics-based feature extraction and selection for the prediction of lung cancer survival," which you have submitted for peer review to the International Journal of Molecular Sciences (IJMS).
After conducting a comprehensive review, I would like to share the following comments with you:
Comment 1: Lack of Novelty
In the rapidly evolving field of cancer research, particularly in the context of TCGA data analysis, it is essential to contribute novel and significant findings. Unfortunately, based on my assessment of your manuscript, I did not identify a sufficient level of novelty in your proposed model and methodology. The research appears to align with the broader trends in TCGA-based studies without introducing unique elements or innovative approaches.
Given the current landscape where numerous papers are published monthly on TCGA data, it is crucial to distinguish your work by offering a novel perspective, method, or insight. I recommend reconsidering the focus and methodology to enhance the uniqueness and impact of your research.
Comment 2: Consideration of Invasive Markers
The choice to use invasive markers for cancer vs non-cancerous cell classification warrants further clarification. In the context of established diagnostic methods such as biopsy, PET scanning, MRI, and CT scanning, it would be beneficial to explain the necessity for incorporating invasive markers alongside these well-proven techniques. Addressing this point would help readers understand the rationale behind employing genomics and machine learning in conjunction with existing diagnostic tools.
While genomics and machine learning bring their own challenges, providing a compelling justification for their integration will strengthen the impact of your research.
Comment 3: Methodological Robustness
One aspect that stood out during my review is the perceived lack of robustness in the methodology. It appears that normalization and appropriate weighting of the dataset were not explicitly addressed during the development of the machine learning model. This could potentially lead to issues such as overfitting, especially when dealing with diverse omics data.
For instance, the use of microRNA expression data might yield suboptimal results due to its generally lower expression levels compared to mRNA. I recommend revisiting the normalization procedures and incorporating proper weighting strategies to enhance the model's robustness and reduce the risk of overfitting.
Best
Author Response
Thank you for providing the review. Below are our detailed answers to each of the 3 raised concerns:
-
We expanded the discussion to highlight the novelties of the feature selection and extraction strategy, and how the survival classification results that we obtained surpass those previously reported. In our opinion, the presented study is novel both in terms of the methodology and data utilized, for the following reasons:
- Our work presents the first comparison of feature extraction methods applicable to the genomic data, by assessing their impact on classification performance. None of the previous articles that utilize machine learning for the prediction of lung cancer survival utilized this strategy, including all articles on this subject listed in the review by Gao Y et al. 2020; doi:10.21037/jtd-2019-itm-013.
- Our work provides a comprehensive evaluation of various omic datasets for the prediction of lung cancer survival, using 3 distinct methods: Lasso, Boruta and This is the first such analysis that provides insights into the contribution of each dataset into the model creation.
- This is also first such study that uses an independent dataset for results validation. The typical workflow utilizes a hold out set derived from the TCGA data while we utilized a new dataset obtained in the CPTAC-3 project to validate our findings.
- To address concerns of overfitting and information leakage, we utilized nested cross-validation, to evaluate the selected features. Both of those problems are often overlooked in previous articles on this subject. To allows for a deeper understanding of the impact each feature has on the model's predictions our study incorporates SHAP (SHapley Additive exPlanations) to elucidate variable contributions, which to our knowledge was not previously utilized for similar classification tasks on the lung cancer data.
-
We agree that the utilization of molecular data for cancer vs non-cancerous cell classification is in most cases unjustified, given the high price of such studies and long time needed to derive the conclusions, compared to the traditional methods listed in the comment. However, our study utilizes classification not for this particular purpose, but rather to predict patient survival, which cannot be successfully carried out with the limited information provided by traditional techniques. One of the goals of our study is to assess the usefulness of the methods used to obtain the high throughput molecular level data, and while we do not compare the outcomes to the established diagnostic methods, we firmly believe that exploring the potential of the molecular level techniques is essential for advancing medical research and enhancing our understanding of cancer. This information will be invaluable in the forthcoming era when high throughput molecular characterization of the tumor will be a routine procedure, however, not for the purpose of identifying the cancer but rather selecting the optimal treatment strategy that would maximize the patient survival.
-
The dependence of the feature selection technique on the measurement scale was one of the major concerns addressed in this project. The feature selection methodology based on Boruta ranking, which was used in the multi-omic comparison, does not show any bias associated with the measurement scale. Boruta relies on a random forest, which is basically an ensemble of decision trees. Its advantage lies in the fact that decision trees do not require scaling, as choosing a split on one scale is equivalent to selecting a re-scaled split on an alternative scale.
This is supported by the results shown in Fig. 4 which highlights the selected features that include both methylation levels with values between 0 and 1 and gene expression levels, with values measured in thousands (log transformation is also not needed here). We discuss this problem at the beginning of section 2.4, where a short sentence was added to highlight the scale independence of the selected method: “Boruta is based on the random forests algorithm, which is an ensemble of decision trees that are scale independent. For this reason various -omic datasets, that utilize significantly distinct measurement scales, can be combined without any additional transformations.”
Reviewer 3 Report
Comments and Suggestions for Authors
Here, the authors report a study on using multiomics data and machine learning methods to predict lung cancer survival. The authors use two lung adenocarcinoma datasets from TCGA and CPTAC-3 projects, and compare various feature extraction and selection techniques. They find that methylation, gene expression, and mutation data are the most relevant for survival prediction, and that gene set aggregation is the most effective feature extraction method. They also identify 32 molecular features that can achieve high accuracy in survival prediction on both datasets.
Major points:
The article provides a comprehensive review of the existing literature and methods on lung cancer prognosis and multiomics analysis.
The article uses two independent datasets to validate the robustness of the selected features and the predictive model.
The article employs various feature extraction techniques to deal with the challenges of sparse and high-dimensional data, and compares their performance using objective metrics.
Minor points:
The article does not discuss the biological or clinical implications of the selected features and their association with lung cancer survival. This should be addressed in the discussion.
The article does not address the potential confounding factors or biases that may affect the survival prediction, such as age, gender, smoking status, treatment, or comorbidities. This could be addressed in the discussion.
The article does not compare the performance of the proposed model with other existing models or benchmarks for lung cancer survival prediction. This should be discussed in a last paragraph.
Comments on the Quality of English Languagesome typos
Author Response
Thank you for providing this insightful review. Below are our detailed answers to each of the 3 suggestions:
"The article does not discuss the biological or clinical implications of the selected features and their association with lung cancer survival. This should be addressed in the discussion."
The first paragraph of the discussion section has been expanded to provide the answer to this remark.
"The article does not address the potential confounding factors or biases that may affect the survival prediction, such as age, gender, smoking status, treatment, or comorbidities. This could be addressed in the discussion."
As the work focused on –omics based features, other potential confounding factors or biases that may affect the survival prediction have not been included in our analysis. However, the issue raised by the reviewer is very important. Therefore, we have added a separate paragraph devoted to research in this area to the Discussion section.
"The article does not compare the performance of the proposed model with other existing models or benchmarks for lung cancer survival prediction. This should be discussed in a last paragraph."
An additional paragraph in the Discussion section, added following the reviewer’s suggestion, addresses this issue.
Reviewer 4 Report
Comments and Suggestions for Authors
According to the authors, the aim of this article is to address the challenges associated with feature extraction and selection by exploring the potential of machine learning in conjunction with multi omics data to improve the prediction of lung cancer survival.
In this sense, the primary goal was to extract and select the most useful molecular features for the prediction: two distinct lung adenocarcinoma multiomics dataset were obtained from the Cancer Genom Atlas (TCGA) and Clinical Proteomic Tumor Analysis 3 (CTPAC).
The study utilized five omics data (somatic mutation, gene expression levels from the total RNA-Seq, microRNA expression levels, methylation levels an CNV) and various feature extraction techniques (Region overlap aggregation, ; Gene level aggregation Gene st aggregation, mutation statistics, mutation signatures, PCA) which were tested using two approaches : lasso variable and Boruta feature ranking.
On the one hand, the evaluation with the 10-fold lasso regression show that gene set aggregation is the best features extraction methods with an AUC statistics of 0.75 and 0.7 of 25 mutation and CNV feature. In addition, the prediction capabilities of each dataset for LUAD patient survival for 2-years and 5-years show that in both case, methylation and mRNA studies provide the most useful prediction allowing achieving AUC exceeding 0.8 and mutation data shows similarly high AUC, though only in the case of 2-years survival. These results are obtained by combination 10-fold lasso regression and gene set aggregation.
On the other hand, using Boruta analysis, the experiment shows that 40 features originating from the methylation and mRNA dataset were confirmed to be high performance score with one of CNV classified as tentative.
Both of the two approaches confirmed that methylation and mRNA are the best features. However, the TCGA-trained classification performed poorly on the CPTCA-3 set, due to significant differences between the two methods used to obtain the datasets and differences in the preprocessing pipeline.
General comments:
The strategy of the author is very well defined, and the results are well organized and presented.
According to the overall quality of this study, I consider this manuscript as acceptable for publication in IJMS
Author Response
Thank you for providing this insightful review of our manuscript.
Round 2
Reviewer 2 Report
Comments and Suggestions for Authors
The authors have not addressed the raised questions and comments from the previous revision. There appears to be a potential deficiency in the methodology section, with several techniques missing. This could result in challenges for replication given the methods chosen by the authors. In the previous revision, I raised concerns about normalization, particularly since the authors integrated various genomics data such as mRNA, miRNA, and proteomics data without applying any normalization methods. Furthermore, there is a lack of innovative methods for dataset classification. While TCGA data is commonly used for cancer classification predictions, there remains a gap in genomics-based biomarkers for cancer classification and survival predictions. There are lots of room and scopes to improve this manuscript and Based on these considerations, I would not recommend this manuscript for publication.
